# Interdisciplinary Perspectives on Restraint Use in Aged Care

**DOI:** 10.3390/ijerph182111022

**Published:** 2021-10-20

**Authors:** Juanita Breen, Barbara C. Wimmer, Chloé C.H. Smit, Helen Courtney-Pratt, Katherine Lawler, Katharine Salmon, Andrea Price, Lynette R. Goldberg

**Affiliations:** 1Wicking Dementia Research and Education Centre, College of Health and Medicine, University of Tasmania, Hobart, TAS 7000, Australia; chloechsmit@gmail.com (C.C.H.S.); Helen.courtneypratt@utas.edu.au (H.C.-P.); Katherine.lawler@utas.edu.au (K.L.); Katharine.salmon@utas.edu.au (K.S.); andrea.price@utas.edu.au (A.P.); lyn.goldberg@utas.edu.au (L.R.G.); 2School of Pharmacy and Pharmacology, College of Health and Medicine, University of Tasmania, Hobart, TAS 7000, Australia; barbara.wimmer@utas.edu.au

**Keywords:** restraint, restrictive practice, chemical restraint, physical restraint, psychotropic, residential aged care, long term aged care, community aged care, nursing home, home care, day care

## Abstract

Restraint use in Australian residential aged care has been highlighted by the media, and investigated by researchers, government and advocacy bodies. In 2018, the Royal Commission into Aged Care selected ‘Restraint’ as a key focus of inquiry. Subsequently, Federal legislation was passed to ensure restraint is only used in residential aged care services as the ‘last resort’. To inform and develop Government educational resources, we conducted qualitative research to gain greater understanding of the experiences and attitudes of aged care stakeholders around restraint practice. Semi-structured interviews were held with 28 participants, comprising nurses, care staff, physicians, physiotherapists, pharmacists and relatives. Two focus groups were also conducted to ascertain the views of residential and community aged care senior management staff. Data were thematically analyzed using a pragmatic approach of inductive and deductive coding and theme development. Five themes were identified during the study: 1. Understanding of restraint; 2. Support for legislation; 3. Restraint-free environments are not possible; 4. Low-level restraint; 5. Restraint in the community is uncharted. Although most staff, health practitioners and relatives have a basic understanding of restraint, more education is needed at a conceptual level to enable them to identify and avoid restraint practice, particularly ‘low-level’ forms and chemical restraint. There was strong support for the new restraint regulations, but most interviewees admitted they were unsure what the legislation entailed. With regards to resources, stakeholders wanted recognition that there were times when restraint was necessary and advice on what to do in these situations, as opposed to unrealistic aspirations for restraint-free care. Stakeholders reported greater oversight of restraint in residential aged care but specified that community restraint use was largely unknown. Research is needed to investigate the extent and types of restraint practice in community aged care.

## 1. Introduction

Over the past decade, there has been increased scrutiny on high rates of psychotropic use and restraint practice in Australian residential aged care from researchers [1,2,3,4], the media [5], policymakers [6,7], human rights [8] and advocacy groups [9]. This attention led to restraint being highlighted as a key area of focus for the Royal Commission into Aged Care Quality and Safety which commenced in 2018 [10]. Further, the Aged Care Act was amended in 2019 to include Australia’s first legislation regulating the use of restraint in residential aged care. From 1 July 2019, aged care providers have explicit obligations in relation to restraint use [11]. The use of restraint must be the strategy of last resort after rigorous assessment and other non-restraint approaches have been trialed. When judged as appropriate, the use of restraint must be the least restrictive form only after informed consent is gained. Moreover, all use must be monitored and reviewed on a regular basis [11].

‘Restraint’ is defined by the Australian Medical Association (AMA) as ‘*a device or medication that is used for the purpose of restricting the movement and/or behavior of a person*’ [12]. The use of restraint in people receiving aged care is often justified based on reducing risk or preventing harm to the person or others [3]. Yet, restraint use is associated with detrimental consequences, including cognitive decline, increased falls, pressure injuries, lessened activities of daily living (ADLs) and death [3,4,13,14,15]. Despite these adverse effects, restraint is used commonly in residential aged care. A recent systematic review and meta-analysis cited the average prevalence over the last two decades of physical restraint as a third (33%) of all residents and chemical restraint use as 32% of residents [16].

Ideally, aged care providers, wherever situated, should strive for a restraint-free environment. However, in practice, it is often difficult to balance risk management with the promotion of autonomy for older people needing care. Similarly, it can be challenging to provide a safe environment but at the same time enhance a person’s quality of life [17]. There will be situations when difficult decisions regarding restraint need to be made. Recognizing this, in 2012, the Australian Department of Health developed a set of resources, the ‘Decision-Making Tools’ (DMT), to guide providers, nursing and care staff, care-givers and, wherever possible, relatives and residents, to make informed decisions about restraint [18,19]. At the start of 2020, our research group was commissioned to update these resources to align with legislative changes and contemporary best practice.

To inform and develop clear, practical and influential resources we conducted qualitative research aimed to explore the attitudes, beliefs and experiences of a diverse group of interdisciplinary stakeholders towards restraint use in aged care. The current interpretations of what constitutes ‘restraint’ were scoped, along with views on amendments to the Australian Aged Care legislation relating to restraint [9].

## 2. Materials and Methods

### 2.1. Design and Sampling

Qualitative research can be defined as the study of the nature of phenomena and strives to understand why something is, or is not, observed [20]. To achieve our research aim we conducted interviews with a variety of care providers and relatives and two dedicated focus groups with management staff. The triangulation of interview and focus group qualitative data was intended to achieve an in-depth understanding of restraint practice, incorporating both individual perspectives and views of homogeneous groups with relevant expertise and experience [20].

Participants of the interviews and focus groups were purposely selected to represent the key stakeholders involved when restraint is proposed and used in aged care. The semi-structured interviews were held with health practitioners working within, or those with relatives living in aged care settings, including residential, community and day care. The first focus group included senior nurse managers and clinical directors based in residential aged care; the second focus group was comprised of community care managers.

### 2.2. Recruitment

For the semi-structured interviews, we recruited participants working in various roles and aged care settings to capture a wide range of interdisciplinary health practitioner perspectives, including registered nurses (RNs), enrolled nurses (ENs), personal care assistants (PCAs), physicians, physiotherapists and pharmacists. We also sought to obtain viewpoints from relatives of people receiving aged care. Most of the interviews were held in Hobart, Tasmania, where the research was based, but participants from other Australian States were also sought. Potential participants were identified by all members of the interdisciplinary research team across their professional networks, e-mailed an information sheet and invited to take part. Those who responded were phoned by J.B. or C.S. who outlined the study and arranged an interview after gaining verbal consent. Written consent was obtained before each interview and focus group.

The focus group participants were recruited by J.B. who sent emails to potential candidate aged care home provider groups and community care providers inviting them to be part of the study. The focus groups were initially intended to be held before the interviews; however, due to workload and uncertainty associated with COVID-19 they were conducted after the majority of the semi-structured interviews were completed. Both focus groups were moderated by J.B. and conducted remotely via video internet platforms.

Approval for this research was obtained from the Tasmanian Health and Medical Human Research Ethics Committee (ID: 20044). As part of the ethics approval, all identifying information was removed from interview data and details of participants were kept confidential. All participants were assured that they could withdraw from the interview or the study at any time. The interviews were conducted between 23 April 2020 and 1 June 2020 and the two focus groups were held in the final week of May 2020. After their interview or focus group, each participant was offered a $100 book voucher to compensate them for the prereading and their interview time. A third of interviewees declined this incentive.

### 2.3. Qualitative Semi-Structured Interview and Focus Group Process

Prior to the semi-structured interviews, all participants were sent a link to the 2012 DMT resource and told they would be asked for their opinions on this resource during their interview [18,19]. Originally, we planned to conduct most of the semi-structured interviews face-to-face; however, due to COVID-19 restrictions, all interviews were conducted by J.B. and C.S. remotely through internet video-meeting platforms or by phone.

The interviews were conducted using an interview guide (see Table 1) developed by all members of the research team which consisted of three pharmacists, two nurses, a physiotherapist, physician and a speech pathologist. Standard demographic questions were followed by a series of closed and open-ended questions, which were adapted as the study progressed and new areas of enquiry emerged [20]. Participants were free to express their views and experiences and diverge from the interview guide. Likewise, the interviewers were free to ask additional questions or omit questions when not considered relevant. During the interview, participants were asked to recall a case where restraints had been proposed or used, so their real-life experiences were described.

As with the individual interviews, each focus group participant was given a link to the 2012 DMT resource [18,19] to enable content review and completed an on-line demographic questionnaire. A topic guide based on the semi-structured interview guide (see Table 1) was customized for residential aged care or community home care for the first and second focus group, respectively. The Focus Group moderator (J.B.) began each session with an introduction to encourage an open environment for participants to share their opinions and experiences [20]. The moderator then initiated group discussion using open-ended questions from the topic guide and ensured that each participant was given several opportunities to speak during the session.

The interviews and focus groups were recorded, stored confidentially and transcribed verbatim by a professional transcription company. Returned transcripts were anonymized and missing/unclear words clarified by C.S. and J.B. by listening to the original recording. Both focus group transcripts and a sample of semi-structured interview transcripts were checked by J.B. and C.S. to verify the accuracy of the transcription. Interview participants were also offered the opportunity to check and amend their transcribed interviews. Nineteen participants were sent transcripts and three made slight amendments. All interview data files were uploaded onto NVivo 12 software for analyses [21].

### 2.4. Data Analysis

Data were thematically analyzed using a pragmatic approach in which the most appropriate research methods were chosen to investigate the topic as opposed to a single paradigm based on a philosophical doctrine [22]. The ‘Framework Method’ [23], often used in applied health care research, was used to answer the research question about what constitutes ‘restraint’. A bubble plot was used to visually represent the frequency of each theme, along with their conceptualized interconnections, creating a framework of these elements grouping codes into sub-themes [23]. Qualitative analysis was undertaken using Braun and Clarke’s six-step process of thematic analysis which involved data familiarization; interim code generation; seeking themes; reviewing themes; defining and naming themes; write-up [24].

To start, three authors (J.B., L.G. and B.W.) familiarized themselves with the semi-structured interviews, made notes and discussed findings. Likewise, J.B. and B.W. read through the focus group transcripts several times and met to review findings. Then, data were independently coded by J.B. and B.W. using a hybrid inductive and deductive approach, with emerging themes hierarchically coded utilizing the NVivo 12 platform [21]. The data were organized into themes and sub-themes, mapped, and interpreted. As themes were identified, they were cross-checked and debated. Any differences in interpretation were resolved by discussion and adjusted until consensus was reached. Exemplar quotes supporting each theme were captured by J.B and B.W.

## 3. Results

### 3.1. Participant Demographics

Participant demographics for the semi-structured interviews are listed in Table 2. A total of 28 participants were interviewed over a six-week period from April to June 2020. Most were females (*n* = 22) and 6 were males, with ages ranging between 28–68 years. Ten interviewees were RNs (4 were also clinical care managers), 5 were PCAs and 1 EN. The other participants included 3 physicians, 3 physiotherapists, 3 pharmacists and 3 relatives. The majority (*n* = 16) of health practitioners interviewed worked in residential aged care, 4 worked exclusively within community care and 2 in day-care centers. Three of the participants, all physicians, worked in both residential and community aged care settings. The majority of participants (*n* = 15) had worked in aged care for ten years or longer. Of the three relatives, one had a sibling living in residential care, one had a parent receiving community care and the final relative’s parent attended a day-care and respite center. Sixteen of the participants lived in Tasmania and 12 were based elsewhere in Australia. The semi-structured interviews lasted between 35 to 85 min.

Demographic data for each focus group are as follows: Six residential care managers, all RNs, participated in the first group (FG Residential). Four were based in Tasmania and two in Victoria, all were female and aged between 48 and 64 years. For the second focus group (FG Community), three community managers participated (two occupational therapists and an RN). All were female and aged between 42 to 53 years. Two other community care managers from Tasmania had verbally agreed to participate in the second focus group but withdrew without explanation on the day of the meeting. The residential care focus group ran for 87 min and the community focus group for 65 min.

### 3.2. Qualitative Themes

Following analysis, five themes relating to restraint practice in aged care settings were identified. They were: “understanding of restraint”, “support for legislation”, “a restraint-free environment is not achievable”, “low-level restraint” and “community restraint use is uncharted”. These themes are described further below.

#### 3.2.1. Theme 1: Understanding of Restraint

All the participants, except two, were able to define what restraint meant to them. Several participants provided more than one interpretation. One of the physicians refused to give a definition, claiming that to do so was a “*meaningless circular pursuit”*. Over half of those interviewed defined restraint under the sub-theme of ‘limiting what people do’ stating that restraint involved stopping people doing things, restricted their movement or impeded their freedom.



*“It is the act of stopping someone from doing something they want to do. Whatever they want to do, be it a decision or action.”*
Physio 1


The next most common sub-theme is related to ensuring the safety of the older person or others. Notably, those defining restraint as needed for safety or to reduce harm were predominantly RNs working in residential care. A few participants, all pharmacists, defined restraint as a means to control a person’s behavior. Several interviewees defined restraint more broadly as also impeding the ability to make choices.



*“It’s about restricting movement, restricting rights, anything about the person’s ability to retain their independence or choice.”*
FG Residential


Three participants stated that restraint altered the mind of a person or the way they thought. Finally, all the PCAs and one of the relatives defined the term in literal terms as either ‘physical’ or ‘chemical’ forms of restraint.

The codes and sub-themes relating to the overriding theme of ‘understanding of restraint’ are presented as a bubble plot below (Figure 1). A summative approach was used to calculate the total number of definitions including a coded element. The size of the bubble is proportional to the number of definitions coded to each sub-theme [23].

Many participants felt that definitions of restraint varied widely depending on the aged care organization, the aged care setting and between staff working within an organization.



*“I think a lot of it comes down to sometimes how people define it, and obviously that changes hugely. Even in one facility, you talk to maybe the manager and they say one thing, and then the RN thinks something different, and then someone else thinks something different; so it can be quite confusing.”*
Pharm 2


Several participants mentioned inconsistencies in restraint definitions used in different government publications, the aged care sector and the National Disability Insurance Scheme (NDIS). This lack of clarity and consistency about what restraint means confuses staff and other health practitioners working within aged care.



*“In the residential context it’s even more complicated, because the national quality indicator program defines ‘restraint’ differently to the Legislation. And in providers where there’s mixed circumstances like ours, where potentially you could have staff providing support to community and residential aged care, it’s diabolical….. we’ve also got NDIS consumers in the community where the restrictive practices obligations are different again.”*
FG Community


#### 3.2.2. Theme 2: Support for Legislation

Participants were very supportive of the new restraint legislation that had been introduced for residential aged care, claiming that it had heightened awareness, made staff seek alternative strategies, ensured greater accountability and had already impacted use.



*“I think it’s great. It’s reined people in. It’s made everybody think about what we’re doing as opposed to just this person is disruptive on the evening shift, we don’t have time to deal with this so let’s just give him something to shut him up. Because that’s what was happening.”*
RN 9


Some participants felt the legislation had directly enhanced interprofessional collaboration, particularly around the use and review of chemical restraint.



*“It has made us focus on the chemical restraints a lot more, and we’ve had a lot more conversations, with General Practitioners (GPs), around ceasing, than we would have had before.”*
RN 8


Although most people interviewed were supportive of the tightened regulation many admitted that they were not entirely sure what it involved. Several nursing staff said the legislative changes had imposed additional workload, such as increased documentation and ensuring informed consent had been gained. Yet, despite increased obligations on providers, most felt the additional reporting and safeguards were worth it to reduce restraint practice.



*“It’s a massive pain, but I think it needed to be done and I’d rather them go ridiculous and way over-report and me have to deal with the paperwork nightmare for the next two years and then slowly reduce it and actually catch out some of the people that were doing the wrong thing.”*
RN 1


#### 3.2.3. Theme 3: Restraint-Free Environments Not Possible

All participants were asked to read the DMT restraint resources [18,19] which incorporate the title: ‘supporting a restraint-free environment’. Yet most queried the feasibility of ‘restraint-free’ practice; expressing the view that locked doors and gates, both forms of environmental restraint, were crucial to have in aged care, particularly when many clients were highly cognitively impaired. If locks were categorized as restraint use, then using restraint in most settings was unavoidable.



*“I don’t know that restraint-free practice is—I don’t know that it’s possible. I mean, we’re talking about me, here. Restraint-free means my door’s open. It’s not ideal. It’s not safe, it’s not possible. Well, it’s possible, I can do it, but what do I say? Well, he got run over yesterday, told him he shouldn’t have gone out the gate.”*
PCA day care 1


Similarly, many participants expressed the opinion that chemical restraint could never be completely eliminated but instead it was more important to ensure they were used appropriately when prescribed.



*“I don’t believe it’s possible to have zero antipsychotics in a facility, or zero psychotropics in a facility, but I definitely think that it should be possible to have only those who have a clear diagnosis, a clear plan, and it’s all monitored.”*
Pharm 3


There was consensus from most stakeholders that restraint was sometimes needed and that the overarching emphasis should be on minimizing use, not to condone all use.



*“I think that needs to be clear from the get-go with recognition that sometimes, restraint is necessary to prevent people from harming themselves or coming to harm or harming other people.”*
Relative 2




*“The care staff know that it’s never going to be a restraint-free environment. To minimize the impact of restraint, you’re minimizing them and having as little as possible.”*
EN


Some of the managers in the residential aged care focus group commented that some homes were catering to the ‘*restraint-free spin’*. They felt that homes voicing they were ‘restraint-free’ demonstrated a limited understanding of what restraint meant and the types of practices it entailed.



*“People want to say, “We don’t have restraint here,” and that’s such a big aspirational target. I think in some instances, there’s real ignorance about what restraint is and what it looks like.”*
FG Residential


#### 3.2.4. Theme 4: Low-Level Restraint

During the residential care focus group and in many of the semi-structured interviews, participants referred to the use of low-level or less obvious forms of restraint. From the residential focus group:



*“All of our facilities say they don’t use any physical restraint, but we found physical restraint: the pushing the chair under the table, the locked doors to outside areas, so a whole lot of things that aren’t seen as hard physical restraint but are definitively restraining.”*
FG Residential Participant 1




*“I see things like call bells that are dropped on the floor or not in positions to allow the person to get assistance.”*
FG Residential Participant 2




*“Even just simple things like leaving a tray-table across a chair that is being used for having a meal or an activity but then not removing it, so the person is free to move around.”*
FG Residential Participant 3


Participants also referred to practices such as tucking in bed sheets tightly to restrict a person’s movement, taking cushions away from deep armchairs and the use of low beds or princess chairs as forms of restraint. One physician mentioned that not accommodating for hearing and/or sight impairment could also be viewed as restraint. Some felt that in most cases this ‘low-level’ restraint use was unintentional and spoke to a lack of awareness and the need for more education on this issue.



*“It’s making people aware of what is considered to be a restraint is really important too. So things like the princess chairs that they use. Or even someone who’s got some sort of incapacity, so they can’t hear everyone, or they can’t see everyone, they’re not able to access help when they want to access help.”*
Physician 1


In contrast, a few participants implied that low-level restraint was used commonly to compensate for a lack of staff or to allow staff to assist other residents.



*“Everyone’s got to try and put other people to bed…one resident can sit there for half an hour after dinner’s finished all by themselves, with the wheelchair locked, because they don’t want them to get away. But no-one’s there to take them back to their room and help them out.”*
PCA 3




*“I suspect, across the board, there’s a lot of, what I would call, low-level restraint to be able to implement the care of anyone in a facility like ours, which is for people with dementia…It’s one of those things where you end up in this argument….”Well, if we can’t do that, we can’t actually implement any care.”*
Physio 2


#### 3.2.5. Theme 5: Community Restraint Use Is Uncharted

Community-based nursing staff and PCAs faced additional barriers when it came to identifying and minimizing restraint. In response to the question, “*do you think that the practice of restraining people occurs commonly in community aged care?*” the community focus group members replied:



*“It’s difficult to judge. It would just be more difficult to gauge in a community setting than it would be in residential aged care because of what we’re in there for and what we’re not caring for.”*
FG Community Participant 1




*“That’s right. And certainly, Home Care Packages, the majority of them, we don’t actually see what medications they’re on, because we’re not providing clinical care; we’re providing case management and other community services……so it could be a really hidden problem.”*
FG Community Participant 2


Participants working for community service providers said that they had limited control over what happened in a client’s own home. If the family installed a bed-rail they were unable to prevent its use when they were not there. In addition, they stressed that many people with home care packages opted not to use their funding for clinical care, including medication management, meaning that the use of chemical restraint could not be ascertained. Adding to the complexity was the use of multiple care providers by the same client, the lack of home visits made by GPs and the need to manage relationships with certain clients who were resistant to having assistance with care:



*“I think you have to step so carefully with some people in the community. Even if we’ve got concerns, we’ve got to be really careful how we manage that, so we don’t affect the relationship, the provisional relationship with the client.”*
FG Community Participant 3




*“Absolutely. We go into homes and there’s medications all over the floor, and we can’t do anything about that, we just have to report it.”*
FG Community Participant 2


Several community nurses and PCAs stressed that they fostered client independence and encouraged family involvement, rather than let the service ‘*take over*’. Paradoxically, several relatives of older people receiving community aged care reported situations where they were not consulted when restraint was proposed and subsequently used:



*“The community nurse called Dad’s GP and to her credit, the GP was reluctant to put Dad on this antipsychotic but everybody else was pushing for it, so she did write a prescription and they used it and then they let me know by email after it was done.”*
Relative 3



Interviewer: *“So, they didn’t ask your permission at all, it was just more informing you that it happened?”*
*“That’s exactly it.”*
Relative 3


All three relatives interviewed agreed there is was a strong need for those receiving community-based care and their families to receive guidance on restraint practice.

## 4. Discussion

This study provides insight into the attitudes, beliefs and experiences of a diverse group of interdisciplinary stakeholders towards restraint use in aged care. ‘Interdisciplinary’ is a term used to describe healthcare practitioners from different professional disciplines who work together to manage the care needs of a person. Whenever possible the person and their family should be an integral part of this group [25]. Our interdisciplinary stakeholders were nursing staff, PCAs, physicians, physiotherapists and pharmacists, as well as relatives of people receiving aged care. Aged care clients, also known as consumers or residents, were not directly involved in this research due to ethical considerations around capacity to consent, alongside restrictions and uncertainty associated with COVID-19. However, a group of Australian researchers was able to interview community-based older people about restraint, reporting that they were conscious of this issue and concerned about being on the receiving end of such practice [26]. Those interviewed were most averse to the use of physical restraint and sedation, which were perceived to have the greatest impact concerning limiting choice and self-expression {26].

Several recent systematic reviews have reported that definitions of restraint in the research literature are highly variable, with interpretations differing according to country, aged care setting, and the timeframe in which studies were conducted [3,27,28]. When we asked our participants to define what restraint meant to them we noted similar differences in understanding. The definitions given by our stakeholders tended to vary according to the professional background of the participant; for example, all the physiotherapists defined restraint as restricting the movement of a person. Likewise, all three pharmacist participants referred to restraint as a way of managing behavior. This would be expected given physiotherapists specialize in movement and pharmacists provide advice on medication that affects mood and behavior. Yet, in spite of a degree of professional variation in interpretation, more than half of the participants defined restraint as ‘limiting what a person can do’ with regards to how they act, move and their overall freedom. This broad definition aligns with the 2019 legislative definition of restraint as ‘*any practice, device or action that interferes with a consumer’s ability to make a decision or restricts a consumer’s free movement*’ [11] and provides some indication that most people in our study conceptualize restraint in line with the new legislation.

The second most common definition of restraint cited by participants, predominantly RNs, was that restraint ensures the safety of older people receiving care and reduces the risk of them harming themselves or others. This is not, in essence, a definition but rather provides justification for why restraint is used. The rationalization for restraint; ‘under the premise of risk minimization and prevention of harm to self or others’, has been reported in Australia as far back as 2005 [28]. It has also been reported in several government enquiries conducted about Oakden, an older person’s mental health service in Adelaide, South Australia [29], despite a lack of evidence that restraint, either physical or chemical, protects residents against injuries and falls [3,14,15,28].

Participants were far more likely to define restraint as restricting movement and actions than to control behavior, an aspect which relates to the definition of chemical restraint as ‘*a practice or intervention that involves the use of medication for the primary purpose of influencing a care recipient’s behavior*’ [30]. This may speak to the difficulty of determining if a psychotropic medication is used for a medical or mental health condition as opposed to influencing behavior. Some older people receiving care may present with a combination of behavioral symptoms and mental illness. The reluctance to define chemical restraint was also observed in a recent systematic review examining the prevalence of restraint practice in residential aged care [16]. The authors of this review could locate only four studies that provided a definition for chemical restraint, compared to 51 studies that explicitly defined physical restraint [16]. The mental health sector has also found it challenging to define chemical restraint, acknowledging ‘*use remains controversial with different understandings of what it is and its role in care*’ [31]. It appears more education is needed around what chemical restraint entails in the aged care sector so that it is identified, and when proposed and/or used, practice accords with the requirements set out in the Aged Care Legislation [11] and the Aged Care Quality Standards [32].

Evidence is mounting that legislation appears to be one of the most effective approaches to reduce restraint practice for older people receiving residential aged care. Countries that have introduced legislation in response to high rates of restraint, including the USA and Canada, have subsequently reported significant reductions in use [16,33,34]. One consistent finding in our study was the high level of support for the recent legislative amendments that have been introduced around restraint use in Australian residential aged care [11]. Although some participants were frustrated with the increased reporting associated with the new Legislation, they agreed that documentation and enhanced oversight was necessary to stamp out poor practice and reduce reliance on restraint use. Interestingly, although highly supportive of enhanced regulation, many participants admitted they did not know the specifics of the legislative amendments relating to restraint, pinpointing a need for additional information and training for staff, healthcare practitioners and informal caregivers.

Another common theme raised in this research relates to the concept of ensuring a ‘restraint-free environment’. The 2012 DMT resources state: ‘*with a restraint free approach, the use of any restraint must always be the last resort*’ [18,19]. Some of the participants expressed the view that in literal terms, ‘restraint-free’ meant that restraint should never be used, as opposed to being permitted in certain situations. As with restraint, the meaning of a ‘restraint-free environment’ appears to vary in different settings and between countries. For instance, a group of researchers based in a hospital in the USA claimed they were able to achieve a ‘restraint-free environment’ in their delirium unit [35]. However, they specifically defined ‘restraint-free’ as meaning no physical restraint. Antipsychotic and benzodiazepine use was still permitted and there was no reference to the use or absence of environmental restraint (i.e., locked doors, keypads) [35]. Many of our participants felt that a completely restraint-free environment was unattainable especially when providers require gates to be locked or keypads used to prevent people with severe dementia leaving services unattended. Others stressed that it was important to acknowledge there were certain times when restraint was needed but instead there should be significantly more emphasis on ensuring use was appropriate and that legislative obligations were followed. To circumvent confusion and skepticism around the term ‘restraint-free environment’ it may be more appropriate to focus instead on using terms such as ‘minimizing restraint’ or ‘restraint as the last resort’.

‘Low-level’ restraint was another practice over which many participants expressed concern, citing examples such as wedging a wheelchair under a table at mealtimes or placing a walker or care-bell out of reach. The use of these less obvious methods of restraint has also been reported by others [17,36,37]. An ethnological study conducted in Norway, consisting of a mixture of field observation and staff interviews, found that low-level restraint practices were often used to avoid using more overt forms of restraint. These low-level practices also allowed staff to *‘get the care work done’*. Similar to some of our participants, the Norwegian researchers reported that many of the care staff were not aware that certain practices could be construed as restraint [36]. Training the staff in what restraint means within a human rights framework, detailing the practices it entails, and encouraging open discussion around restraint is vital to mitigate this issue.

Participants based in community aged care reported additional challenges with regards to restraint. They reported that due to the ethos of ‘consumer directed care’ in home care, clients now control the types of care and services they receive [38]. This means that for many clients, clinical care is not provided as part of their home care package. Community care nurses and PCAs would often only visit a client’s home briefly so had limited knowledge of the medication they were taking or if they were subject to other forms of restraint. It should be stressed that the new legislative obligations on restraint [11] only apply to residential aged care providers, although the Aged Care Quality Standards [32], which apply in all federally-funded aged care settings, do require providers to minimize restraint and report on restraint as part of governance.

Our health professionals reported that families, in general, were more involved with the care of their loved one when they were living in the community and stressed the need to work collaboratively with them. However, several of our community-based participants recounted experiences where their relatives with dementia were commenced on psychotropic medication for behavior control and they only found out after the medication had been administered. Similar experiences of chemical restraint use without informed consent were recounted in a 2019 Human Rights report [8] and at the Royal Commission [10]. Additional training for health care practitioners around the legal requirement for obtaining informed consent is urgently needed to address this issue. In a recent research study [2], also presented as evidence to the Royal Commission [10], rates of psychotropic use in the community were shown to increase markedly in the year prior to aged care admission, providing an indication that the issue of chemical restraint is not just confined to residential care. More research is needed to gauge the extent of restraint practice, both chemical and other forms, in community aged care, particularly given the rapid expansion of the Home Care sector. Further, the knowledge, experiences and opinions of relatives around restraint use in all aged care settings need to be investigated in greater depth as their voice is rarely captured [39].

Community care nurses and PCAs also reported that they had little influence if the client or their family decided to use devices such as bedrails or low beds, both forms of restraint, as they were only responsible for practices that occurred while they were present providing care. Similar barriers were reported in a Dutch qualitative study of restraint use in community settings that concluded that informal caregivers, especially relatives, have a dominant role in the use of restraint [40]. They also reported that relatives were less aware of the harms associated with restraint use and had limited knowledge of alternative strategies. Due to their findings, the researchers have instituted a training program on restraint for informal caregivers, with promising results to date [40]. Similar education and training may be needed in Australia.

There are several limitations associated with this study. First, we found it challenging recruiting participants based in community aged care. Although we approached several large community care providers in Tasmania, requesting names of potential staff to interview, volunteers were not forthcoming. Two participants from a large Home Care provider also withdrew on the day of their focus group. We theorize that this lack of recruitment was due to several factors, including staff shortages, limited experience with aged care research, and the emerging COVID-19 situation at the time. As there were fewer participants from the community compared with the sample from residential aged care, their experiences and opinions regarding restraint may not be representative of the community aged care sector overall. Second, we need to note that all participants were asked to read a 2012 resource on restraint [18,19] before being interviewed or involved in one of the focus groups. This pre-reading may have increased knowledge and influenced viewpoints around restraint use, meaning that their opinions and familiarity about this topic may not be representative of key stakeholders overall. A final limitation involved the online nature of the interviews and focus groups owing to restrictions imposed at the initial stages of the pandemic. We inevitably experienced technical difficulties with sound and vision with some participants, resulting in several interviews having to be rescheduled, and some conducted solely using mobile phones. Participants may not have been as willing to express their opinion due to the lack of direct personal engagement with the interviewer, and with the lack of interaction with other participants in focus groups. Despite these limitations, we felt that all participants were very accommodating of the situation and that rich and insightful data regarding restraint in aged care settings were obtained.

## 5. Conclusions

This study was conducted to inform and update Government resources on restraint to align with legislative changes and contemporary best practice. The findings suggest that many nursing and care staff, health practitioners and relatives have a broad understanding of what restraint means. However, additional education is needed for these stakeholders on a more conceptual level about what restraint involves, particularly how to recognize and minimize low-level forms and chemical restraint. Likewise, resources need to contain information about the new Aged Care Legislation relating to restraint and what this means for providers and consumers of aged care services. Resources for providers advocating restraint-free environments were considered aspirational. Instead more practical guidance on what to do to prevent and minimize restraint, as well as what to do when restraint is judged appropriate was sought. Research is needed to investigate restraint practice in community aged care as use is largely unknown.

## Figures and Tables

**Figure 1 ijerph-18-11022-f001:**
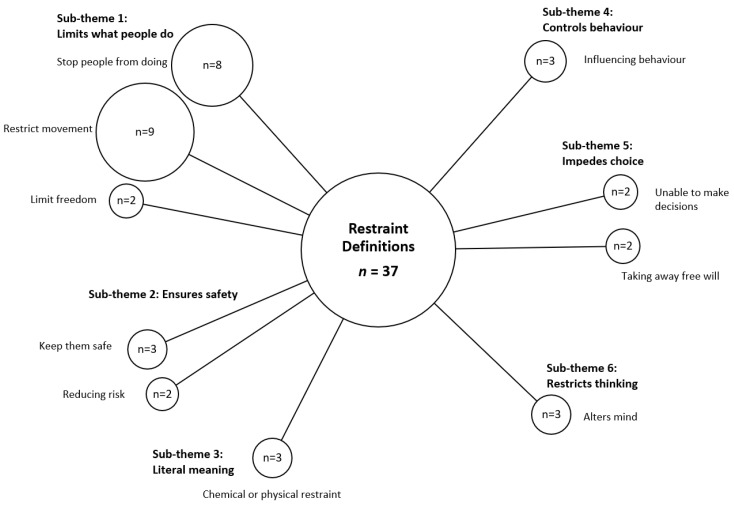
Codes and Sub-themes relating to participants’ understanding of restraint. *n =* number of definitions.

**Table 1 ijerph-18-11022-t001:** Semi-structured interview guide.

Our project involves updating resources for the Aged Care Quality and Safety Commission (ACQSC) which supports aged care providers, health practitioners and caregivers and wherever possible, consumers, to make informed decisions about restraint. To start with can I ask you what the term ‘restraint’ means to you in the context of providing care to older people? In your opinion do most people you work with have the same definition—or do you think that other people have a different definition of what constitutes restraint? Do you think the practice of restraining people occurs commonly in aged care? I sent you a copy of the restraint decision making tool—did you have a read through it? What do you think about the current tool for restraint use? Could you provide an example of a case when restraint might be needed? Could you describe this case for me? (then ask questions to find out things such as what happened? What did the staff do? Was this effective? What else was tried? etc?) With this case, do you think you were supported by the organisation’s policy? Along a similar theme, what are your thoughts about the new restraint amendments to the Aged care Legislation? These are all the questions I have listed—but would you like to add any other comments about this topic?

**Table 2 ijerph-18-11022-t002:** Semi-structured interview participant demographic information.

Participant	Gender	Age	Role	Setting	State	Years in Aged Care
1	F	62	RN 1	RACF	Tas	>10
2	F	57	Physician 1	RACF/Comm	NSW	>10
3	F	42	Pharm 1	RACF	NSW	5–10
4	M	40	Physio 1	RACF	Tas	5–10
5	M	56	Pharm 2	RACF	Vic	>10
6	F	55	Relative 1	Comm	NSW	3
7	F	44	Physician 2	RACF/Comm	Tas	>10
8	F	56	Relative 2	RACF	QLD	2
9	F	50	Physio 2	RACF	WA	>10
10	F	53	PCA 1	Comm	NSW	>10
11	M	31	RN 2	RACF	Tas	1–5
12	F	68	Relative 3	Day Care	Tas	1
13	F	30	RN 3	RACF	Tas	>10
14	M	32	RN 4	Comm	Tas	5–10
15	F	60	RN 5	RACF	Vic	>10
16	F	55	Physio 3	RACF	Vic	5–10
17	F	28	RN 6	RACF	SA	5–10
18	M	55	Pharm 3	RACF	SA	>10
19	F	29	PCA 2	RACF	Tas	1–5
20	F	53	Physician 3	RACF/Comm	Tas	>10
21	F	56	RN 7	RACF	Tas	1–5
22	F	30	RN 8	RACF	Tas	1–5
23	F	45	EN	RACF	Tas	>10
24	M	52	RN 9	Comm	Tas	>10
25	F	28	PCA 3	RACF	Tas	1
26	F	51	RN 10	RACF	Tas	>10
27	F	62	PCA 4	Day Care	Tas	>10
28	F	60	PCA 5	Day Care	QLD	>10

## Data Availability

Full anonymized data supporting reported results are available by contacting the lead author via juanita.breen@utas.edu.au.

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
