# Peer review of "Interdisciplinary Perspectives on Restraint Use in Aged Care"

_ijerph, 2021, doi:10.3390/ijerph182111022_

Round 1

Reviewer 1 Report

Thank you for asking me to review this extremely well written manuscript that reports on an important piece of research that significantly contributes to the field of research. Understanding how an interdisciplinary team views restraint use in aged care and the community and the perceived issues related to achieving a restraint free aged care environment is essential to developing strategies that will go toward achieving this.

There is very little I can find to change in this manuscript – the available literature has been reviewed and summarised well; the methods are clearly described and are appropriate to answer the research question; the results are clearly reported with appropriate quotes to support findings; and the discussion is well developed. My only recommendation would be to reduce some of the sample quotes as some are quite long and repetitive without adding to the discussion. Some points can be summarised and one quote to represent the overall theme supplied.

Congratulations on completing this important research and I look forward to seeing how this information will be used to inform practice.

Author Response

Thank you for your kind and considered review.

Point 1: You suggested that we "reduce some of the sample quotes as some are quite long and repetitive, and do not add to the discussion". You also noted that "some points can be summarised and one quote used to represent the overall theme supplied".

Response: We have now reduced the number of quotes as suggested, removing four and reducing the size of two. This was challenging as we sought for all types of health practitioner voices to be represented. In addition, as focus groups were involved we felt the sequential conversation between participants needed to be documented. We also trimmed down the five theme points.

The manuscript is more succinct because of this.

Thank you.

Reviewer 2 Report

Thank you for the opportunity to review this manuscript.  Overall the authors did an excellent job.  

There are minor general errors such as syntax, spelling or use of abbreviations. For instance, consider revising the sentence from line 52 - 55.  It is too long.   Also, consider revising the word "doctor" throughout out the manuscript to "physician" or "provider".    I recommend to delete the sentence, line 73-74, which defines qualitative research.  I do not believe it is necessary.  

Table 1.  Question B. is not clear to the reader.   Line 192, start of sentence appears misspelled.   The use of GP throughout the document is clarified as "general practitioner" later in the document, see line 352.  I recommend clarifying earlier.  

Although the author's contributions are provided at the end of the manuscript, consider providing background and experience of the researchers.  

The authors described verbal and written consent were obtained prior to each interview.  Consider adding a statement with a description of confidentiality and the voluntary nature of participation.

Author Response

Thanks for your review and thoughtful comments.

 Point 1: You said there were minor general errors such as syntax, spelling or use of abbreviations.

Response: We have edited the manuscript to rectify these errors. We have revised and clarified the abstract sentence from line 52 - 55. It now reads: A recent systematic review and meta-analysis cited the average prevalence over the last two decades of physical restraint as a third (33%) of all residents and chemical restraint use as 32% of residents. We have also revised the word "doctor" throughout out the manuscript to "physician", replacing eight references in total.

 Point 2: Remove the definition of qualitative research. Response: Although we appreciate your suggestion to remove the definition we would prefer to retain this sentence for the following reasons. Firstly, there are various definitions of qualitative research and we wanted to include the definition we used. Second, some readers are not familiar with qualitative research, especially those from a biomedical background or without an academic background (lay people). We hope that this article will be read by a large audience, including these relatives and carers. We feel this justifies the inclusion of the definition. Finally, the definition frames the ensuing method section. Removing it makes the method start rather abruptly.

Point 3. You mentioned that Table 1. Question B is not clear to the reader.

Response: You are correct. We think this sentence may have been altered during the editing process. We have corrected this question now to: 2. In your opinion do most people you work with have the same definition – or do you think that other people have a different definition of what constitutes restraint?

Point 4: You stated in Line 192, that the start of the sentence appears misspelled. 

Response: You are correct in that the sentence could be expressed more clearly so we altered it to: All of the participants, except two, were able to define what restraint meant to them.

Point 5: You noted 'The use of GP throughout the document is clarified as "general practitioner" later in the document, see line 352.  I recommend clarifying earlier.'

Response: There are 4 references to  of GP. We have now moved the definition to after the first mention (line 247).

Point 6. Although the author's contributions are provided at the end of the manuscript, consider providing background and experience of the researchers.

Response: Thanks, we agree but the journal instructions do not permit us to do this at the end. We have included this information now at line 119 in the method section.

Point 7: The authors described verbal and written consent were obtained prior to each interview.  Consider adding a statement with a description of confidentiality and the voluntary nature of participation.

Response: We have added the following to our method at line105: As part of the ethics approval all identifying information was removed from interview data and details of participants kept confidential. All participants were assured that they could withdraw from their interview or the study at any time.

Thank you for your review, especially those relating to providing a more detailed description of the method. This enhances the submission.